# Production of *Arthrospira* (Spirulina) *platensis* Enriched in β-Glucans through Phosphorus Limitation

**Giorgos Markou *** , **Christos Eliopoulos, Anthoula Argyri** and **Dimitrios Arapoglou ***

Institute of Technology of Agricultural Products, Hellenic Agricultural Organization—Demeter,
L. Sof. Venizelou 1, 14123 Lykovrysi, Greece; gmarkou@itap.com.gr (C.E.); anthi.argyri@gmail.com (A.A.)
* Correspondence: markougior@gmail.com (G.M.); dimarap@yahoo.com (D.A.);
  Tel.: +30-2102845940 (G.M. & D.A.)

**Featured Application:** *Arthrospira* **(Spirulina)** *platensis* **enriched in β-glucans could be a potential ingredient for the preparation of novel functional foods.**

**Abstract:** (1) Background: *Arthrospira* (commonly known as Spirulina) is an edible cyanobacterium that is produced worldwide as a food supplement owing to its high nutritional value. *Arthrospira* displays strong potential as an important ingredient in the development of novel functional foods. Polysaccharides from *Arthrospira* are biologically active compounds and hence there is interest in producing biomass rich in carbohydrates. (2) Methods: *A. platensis* was cultivated under different degrees of phosphorus limitation in order to trigger the accumulation of carbohydrates. The biomass was then characterized in terms of its content of α- and β-glucans, total dietary fiber and monosaccharide profile. Fourier-transform infrared spectroscopy (FTIR) was used for the rapid analysis of the main biomass components. (3) Results: Phosphorus limitation resulted in an increase in carbohydrates (from 23% up to 65% dry biomass) of which 4–12% (in relation to the dry biomass) was α-glucans and 20–34% was 1.3:1.6 β-glucans, while 1.4:1.6 β-glucans were not detected. Total dietary fibers ranged from 20–32% (of dry biomass), whereas among the carbohydrates, the predominant monosaccharide was glucose (>95%). FTIR performed well when applied as a prediction tool for the main biomass components. (4) Conclusions: Since β-glucans are of particular interest as biologically active compounds, this study demonstrates that phosphorus-limited *A. platensis* could be a potential ingredient for the development of novel functional foods.

**Keywords:** Spirulina; glucans; nutrient limitation; functional food; polysaccharides; FTIR

## 1. Introduction

Microalgae and cyanobacteria are a very interesting group of photosynthetic microorganisms that have strong potential for use in the food industry as a source of valuable biomolecules (proteins, lipids, pigments, etc.). They have attracted increased interest as cell factories, and as an alternative pathway to traditional practices that employ heterotrophs. As they are photosynthetic microorganisms, they might provide simpler cultivation facilities and easier processes (no sterilization required, low contamination potential, etc.) for the production of biomass and the subsequent biomolecules [1–3]. More recently, microalgae have attracted attention for the production of carbohydrates (polysaccharides) and as a potentially renewable source for various applications in the food and chemical industry [4–8]. Microalgae and cyanobacteria synthesize carbohydrates through photosynthesis and carbon fixation processes. Carbohydrates have diverse structural (cell walls, etc.) and metabolic (energy storage, etc.) roles. The most common carbohydrate types found in microalgae and cyanobacteria are cellulose as structural carbohydrates (in microalgae), and starch (in microalgae) or glycogen (in cyanobacteria) as energy storage molecules [8].

Among the different carbohydrate types, β-glucans are of particular interest due to their bioactive functions, including immunomodulation, antitumor, antioxidant and

antibacterial activities, serum cholesterol and glucose reduction, obesity prevention, etc. [9]. Consequently, β-glucans are considered as a very interesting source for the preparation of functional foods [10,11]. So far, the main sources of β-glucans that have been investigated are cereals, fungi (yeasts, mushrooms, etc.) and bacteria [12]. Similar studies on β-glucans from microalgae and cyanobacteria are scarce [13–15].

*Arthrospira* (commonly known as Spirulina) is an edible cyanobacterium that is produced worldwide as a food supplement owing to its high nutritional value. It is one of the most commercially produced species and is regarded as a very good source of proteins, with a content of more than 60% DW, pigments (phycocyanin and chlorophyll), vitamins, antioxidants, etc. Generally, *Arthrospira* displays strong potential as an important ingredient in the development of novel functional foods that fulfil consumer demand for nutritional and health-beneficial food [16,17]. Regarding the polysaccharides from *Arthrospira*, several studies have demonstrated their biological activity [18–20] and their potential for use in the food industry [4,21]. However, there are no studies known so far that deal with β-glucans produced by *A. platensis*.

An important aspect in regard to the biotechnological application of *A. platensis* is that its carbohydrate content can be significantly increased (from 10–20% up to 50–70%) by triggering the accumulation of carbohydrates through the manipulation of given cultivation parameters, such as temperature, salinity, nutrient availability and light intensity [5,6,22]. Nutrient starvation/limitation (such as nitrogen or phosphorus) of *A. platensis* has been widely studied and proven to be a very effective technique for carbohydrate accumulation [6,23–25] that could also be easily applied in large-scale cultivation systems [22,26]. Nevertheless, there are only a few studies that have focused on the characterization of the carbohydrates produced by nutrient-starved/limited *A. platensis* with regard to nitrogen starvation/limitation [22,27,28], while there is lack of work characterizing the carbohydrates accumulated by the phosphorus starvation/limitation process. The aim of the present study was to optimize the carbohydrate production of *A. platensis* through phosphorus limitation and to characterize the produced biomass in terms of the content of glucan types, dietary fiber, and the monosaccharide profile. Moreover, since the process of phosphorus limitation resulted in the varied biochemical composition of the biomass produced, Fourier-transform infrared spectroscopy was used as a rapid, non-destructive tool for the prediction of the content of the main biomass compounds.

## 2. Materials and Methods

### 2.1. Microorganisms and Cultivation Conditions

The strain used in this study was *Arthrospira platensis* SAG 21.99 obtained from SAG (Sammlung von Algenkulturen der Universität Göttingen). *A. platensis* was grown in modified Zarrouk medium as described in a previous study [25], while phosphorus was supplied in the form of $K_2HPO_4$ in four different concentrations, namely, 1.5, 2, 3 and 4 mg-P/L (and 89 mg-P/L for the control). Cultivation was carried out in closed cylindrical glass photobioreactors with an inner diameter of 76 mm and a working volume of 0.5 L. The cultures were aerated with filtered air provided by a membrane air pump to agitate the cultures. Cultivation was performed in a room with a controlled temperature of 28 °C (±2 °C). Light intensity was 150 μmol $m^{-2}$ $s^{-1}$ (measured by SpectraPen, PSI, Czech Republic), which was provided through a LED panel on one side of the photobioreactors with a photoperiod of 16 h light and 8 h dark (based on preliminary studies). Cultures were carried out in triplicates. The cultures were performed in a semi-continuous mode with a daily feeding rate 20%. The cultures lasted for at least 30 days and the daily harvested biomass was filtered with nylon mesh (50 μm pore size), rinsed with deionized (DI) water and cumulatively stored (combined daily harvested batches) in the freezer (−20 °C), and finally, lyophilized for further analyses. The cultures were monitored frequently (every 3–4 days) through the optical density (OD@750nm) to ensure that the process was steady in terms of biomass concentration. The values of OD did not vary by more than 10%. throughout the cultivation period.

## 2.2. Analytical Methods

### 2.2.1. Biomass Biochemical Composition

Biomass dry weight was measured after filtration with filter paper (pore size 40 μm), being rinsed with DI water to remove the salts from the growth medium and then dried in an oven (65 °C) until constant weight was achieved. Carbohydrates were measured by a modified phenol-sulfuric acid method [29]: briefly, 10 μL of 90% phenol solution were added and mixed with 0.5 mL of cell sample containing 10–50 mg/L carbohydrates, followed by the addition of 1.25 mL of concentrated sulfuric acid (96%). After 30 min, the OD was measured at 485 nm using D-glucose as the standard sugar. Reducing sugars, were determined by the 3.5-dinitrosalicylic acid (DNS) method [30]: 0.25 mL of DNS was added to 0.25 mL of sample and boiled for 5 min in a water bath. Then, 0.44 mL of Rochellet salt and 1 mL of DI water was added and the OD was measured at 540 nm. Lipids were measured by a modification of the sulfo-vanillin method [31] after the extraction of lipids with 2:1:0.2 chloroform:methanol and water. Briefly, 20 μL of extracted sample containing 200–500 mg/L of lipids were incubated in 80 °C to evaporate chloroform and then 0.4 mL of 96% sulfuric acid was added and samples were placed in boiling water for 10 min. Samples were then left at room temperature for 15 min to cool down and 1.0 mL of phosphoric-acid/vanillin solution was added (solution stock was prepared by dissolving 0.12 g of vanillin in 20 mL DI water, and finally, in 80 mL of 85% phosphoric acid). The samples were incubated at 37 °C for 15 min and OD was measured at 530 nm. For the standard curve, corn oil was used. Proteins were assayed according to Lowry et al. [32] after the extraction with 0.5N NaOH: in brief, 1.5 mL of samples were centrifuged, the pelleted biomass was resuspended in 1.5 mL 0.5 N NaOH and then incubated on an agitation heating plate at 100 °C for 20 min. An aliquot of 100 μL of extracted proteins was then added to 100 μL 5% SDS, and supplemented with 1 mL of a solution consisting of 2% $Na_2CO_3$ in 0.1 N NaOH. After 15 min, 100 μL of freshly prepared 1N Folin–Ciocalteu reagent was added and samples were left for 30 min in the dark. The OD was measured at 750 nm using bovine serum albumin as the standard. The final data given are the average of three analytical replicates ($n$ = 3) with the standard deviation.

### 2.2.2. Glucans Determination

Glucans where assayed using enzymatic methods, and more specifically, the commercial products from Megazyme Ltd. (Wicklow, Ireland): (1) β-Glucan Assay Kit (mixed linkages) with the product code: K-BGLU for determining 1.3:1.4-β-D-glucan, and (2) the β-Glucan Assay Kit (Yeast & Mushroom) with product code: K-YBGL for determining 1.3:1.6-β-glucan and α-glucan.

### 2.2.3. High Pressure Liquid Chromatography for Monosaccharides Profile Determination

Monosaccharides were determined by high-pressure liquid chromatography (HPLC) after acid hydrolysis. Briefly, 0.05 g of dried biomass was weighed, placed in a test tube and 10 mL HCl 2M was added. The sample was boiled for 30 min. After boiling, 10 mL of NaOH 2M was added in order to neutralize the mixture. Finally, the sample was filtered and subjected to HPLC analysis. An Agilent Technologies 1100 Series HPLC System with a refractive index detector and a UV/Visible detector was used for determination of sugars. The chromatographic separation of sugars was performed on an Aminex HPX-87H Ion Exclusion, Biorad (300 × 7.8 mm) column with the thermostat set at 50 °C, using a refractive index detector (RID). The determination of organic acids was performed simultaneously in the same conditions by monitoring the UV spectra at 210 nm. The analysis was performed isocratically at a flow rate of 0.7 mL per min and the mobile phase was 0.009 N $H_2SO_4$. The injection volume was 20 μL. The data were processed with the Chemstation Agilent Technologies Software. Duplicate analysis was performed for all samples.

### 2.2.4. Total Dietary Fibers

Total dietary fibers were determined as follows: 200 ± 0.1 mg of dried biomass was weighed and placed into 250 mL beakers in duplicates and 25 mL $H_2O$ was added. Stirring

was used until the test portions were thoroughly hydrated. The beakers were covered with aluminum foil and placed without stirring in an incubator at 37 °C for 90 min. Subsequently, 100 mL 95% ethanol was added to each beaker and left for 1 h at room temperature ($25 \pm 2$ °C). The residue was collected under vacuum in a pre-weighed filter (ashless, No 41, cat No 1441-070). Sample residues were washed 2× with 20 mL of 78% ethanol, 2× with 10 mL of 95% ethanol, and 1x with 10 mL of acetone. Filters containing the residues were dried at 105 °C until constant weight was achieved. Ash was measured after sample combustion at 525 °C for 5 h.

### 2.2.5. Fourier-Transform Infrared Spectroscopy (FTIR) Spectra

FTIR spectra were collected using ZnSe 45° attenuated total reflectance (ATR) through plate crystal, with a HATR sampling accessory on a PerkinElmer Frontier FTIR Spectrometer equipped with a DLaTGS detector with a KBr beam splitter (PerkinElmer Inc., Seer Green, UK). The spectrometer collected spectra over the wavenumber range 4000–650 $cm^{-1}$ and was programmed with PerkinElmer Spectrum v10.4.2 software (Seer Green, UK). Ten scans per measurement were used with a resolution of 4 $cm^{-1}$. The dried samples were placed on the ZnSe 45° ATR crystal and pressed with a gripper to have the best possible contact with the crystal surface. Reference spectra were obtained by collecting an air background (spectrum from the cleaned blank crystal) prior to each sample measurement. At the end of each sampling, the crystal surface was first cleaned with detergent, washed with distilled water, cleaned with ethanol and dried with lint-free tissue. Three (3) replicate FTIR spectra were collected from each of the 3 different batches (biological replicates) of each treatment.

The statistical analysis of the main effects of phosphorus limitation on biomass composition was based on analysis of variance (ANOVA, one-way comparisons), conducted using SigmaPlot 12.0 software (Systat Software, Inc.). Regarding the FTIR model, data analysis was performed using Unscrambler software (version 9.7, CAMO, Norway). The collected FTIR spectral data were background corrected using the standard normal variate (SNV) transformation. Subsequently, spectral data in the range of 1800–650 $cm^{-1}$ were used to build the partial least squares regression (PLS-R) models. The database was partitioned into a training and an external validation dataset. The measurements from the two first batches (biological replicates) of samples were included in the training dataset and measurements from the third batch were included in the external validation set (test set). The normalized FTIR data were subjected to partial least squares (PLS) analysis for the estimation of the main quality parameters of Spirulina content (% DW), i.e., proteins, carbohydrates, lipids, β-glucans, α-glucans, dietary fiber, and glucose. The PLS-R model was evaluated by using a leave-one-out cross validation procedure on the training set only. The number of latent components needed to yield the lowest root mean square error (RMSE) of the cross-validation was evaluated for the modelling and the plot of cross-validation residual variance against the number of latent components was examined in parallel, with up to 20 components included. In the case that the residual variance no longer decreased with additional components, the number of latent components of the first minimum value of residual variance was selected to avoid overfitting. Then, the developed models were validated by applying the test data set. The performance indices for evaluating and comparing the models were the coefficient of determination ($R^2$), the RMSE and the residual prediction deviation (RPD) for the known (observed) values versus validation estimations (predictions) for each parameter.

## 3. Results and Discussion

### 3.1. Biomass Production and Proximate Biochemical Composition

The biomass production of *A. platensis* grown with different phosphorus concentrations is illustrated in Figure 1a. As shown, the increase in the phosphorus provided in the cultures resulted in increased biomass production. This indicates that phosphorus was the growth limiting factor for the cultures containing 1.5, 2 and 3 mg-P/L. When *A. platensis* was cultivated with 4 mg-P/L it produced as much biomass (around 1700 mg/L) as the

control (no statistically significant differences), showing that phosphorus did not affect growth of *A. platensis* at this concentration.

**(a)**

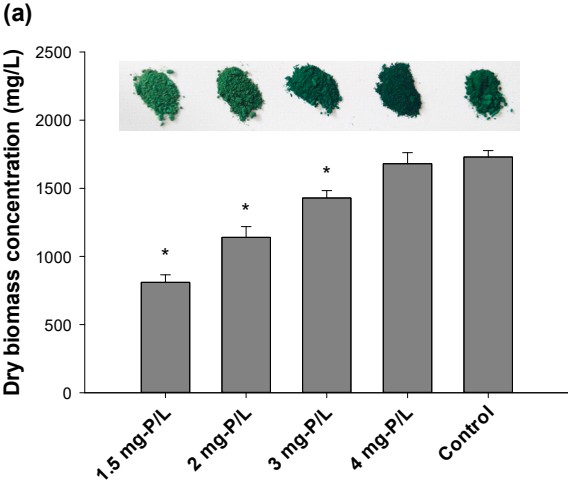

**(b)**

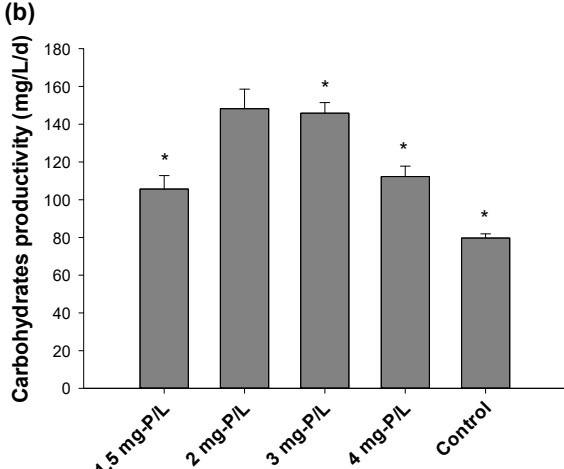

**Figure 1.** (**a**) Biomass production (dry weight) and (**b**) carbohydrate productivity by *Arthrospira platensis* cultivated with different phosphorus concentrations (mg-P/L) in semi-continuous mode (daily feeding rate 20%). Each bar represents the average ± SD of *n* = 3 replicates. The asterisks indicate that there were statistically significant differences between the averages of the sequential cultures.

As shown in Table 1, the proximate biomass composition was significantly affected, in terms of proteins and carbohydrate content, by the amount of phosphorus provided. At the lower P concentrations, there was a strong decrease in protein content (22–25% vs. 60% of the control) and a strong accumulation in total carbohydrates (50–65% vs. 23% of the control) as a consequence of the phosphorus limitation effect [25]. At a phosphorus concentration of 4 mg-P/L, even though the biomass concentration was not affected (Figure 1a), there was a significant difference in the protein and carbohydrate content, indicating that there was still a phosphorus limitation effect compared to the control culture. In terms of carbohydrate productivity (Figure 1b), the best results were obtained with 2 and 3 mg-P/L phosphorus concentration, which gave approximately 145–148 mg/L/d.

**Table 1.** Biochemical composition of *A. platensis* cultivated in semi-continuous mode with different phosphorus concentrations.

| P-Concentration (mg-P/L) | Total Proteins (%$_{DW}$) | Total Carbohydrates (%$_{DW}$) | Total Lipids (%$_{DW}$) |
|---|---|---|---|
| 1.5 | 22.45 ± 3.55 | 65.25 ± 3.88 | 5.56 ± 1.03 |
| 2 | 23.09 ± 1.57 | 65.01 ± 3.24 | 6.68 ± 1.16 |
| 3 | 25.88 ± 1.99 | 51.06 ± 3.24 | 7.98 ± 1.66 |
| 4 | 33.02 ± 2.91 | 33.42 ± 2.67 | 7.38 ± 1.98 |
| 89 (Control) | 59.03 ± 3.61 | 23.04 ± 1.95 | 8.58 ± 1.43 |

One of the main challenges in cultivating microalgae and cyanobacteria under nutrient limited conditions is the trade-off between the accumulation of the target biomolecules triggered by the low availability of the nutrients and the subsequent reduction in biomass production [33]. The results of the present study demonstrate that the process of phosphorus limitation could be optimized in order to cultivate *A. platensis* in long-term cultivation processes by providing phosphorus at amounts that can support cell multiplication while at the same time triggering the down-synthesis of proteins and accumulation of carbohydrates.

### 3.2. Content of Glucans

In order to further characterize the carbohydrates contained in *A. platensis* after phosphorus limitation, two different assay kits were used (see Material and Methods section). The assays revealed that there was an absolute absence of 1.3:1.4-β-D-glucans (not detectable by the assay kit used), while the content of α-, and 1.3:1.6-β-glucans varied significantly (Table 2). The total glucan content displayed an inverse relationship with the phosphorus concentration, i.e., at higher degrees of phosphorus limitation the content of glucans was higher. The same trend was observed for each individual type of glucans measured, namely, α-glucans and 1.3:1.6-β-glucans, with the only exception being that at 4 mg-P/L the content of α-glucans was the same as the control. Since the content of β-(1.3:1.6) and α-glucans showed the same trend, their ratio was calculated and illustrated in Figure 2 as a function with phosphorus availability. As shown, the β/α-glucans ratio was positively related to phosphorus availability, which means that the proportion of α-glucans was higher in the cultures with a higher degree of phosphorus limitation. However, the ratio β/α-glucans in the control culture did not follow this trend, indicating that there was an unbalanced accumulation of α- and β-glucans. Given that the α-glucans content of the cultures with 4 mg-P/L was almost the same as the control cultures, this might suggest that under moderate degrees of P limitation, first β-glucans start to accumulate and then as the degree of P limitation increases the α-glucans also start to increase in an unbalanced pattern of α- and β- glucans. Figure 3 illustrates the proportion of total glucans compared to the total carbohydrates of *A. platensis*. As shown, at higher degrees of phosphorus limitation, the proportion of glucans was lower (around 65–68%) in comparison to the less stressed cultures where the proportion was higher (around 75%) or compared to the control (80%).

**Table 2.** Content of glucans in *A. platensis* cultivated in semi-continuous mode with different phosphorus concentrations. Results represent the average ± SD of *n* = 3 replicates.

| P-Concentration (mg-P/L) | Total Glucans (α + β) (%$_{DW}$) | 1.3:1.6-β-Glucans (%$_{DW}$) | 1.3:1.4-β-Glucans (%$_{DW}$) | α-Glucans (%$_{DW}$) |
|---|---|---|---|---|
| 1.5 | 42.96 | 30.98 ± 1.56 | Not detectable | 11.98 ± 1.55 |
| 2 | 44.56 | 33.62 ± 0.63 | Not detectable | 10.94 ± 0.84 |
| 3 | 35.17 | 27.88 ± 0.16 | Not detectable | 7.29 ± 0.18 |
| 4 | 24.97 | 20.30 ± 1.43 | Not detectable | 4.66 ± 0.07 |
| 89 (Control) | 18.54 | 13.72 ± 0.40 | Not detectable | 4.82 ± 0.57 |

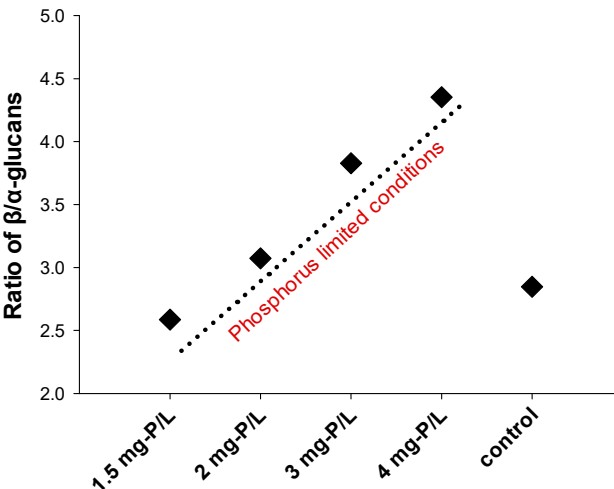

**Figure 2.** Ratio of β- to α-glucans in relationship to the phosphorus concentration (mg-P/L) in cultures of *A. platensis*.

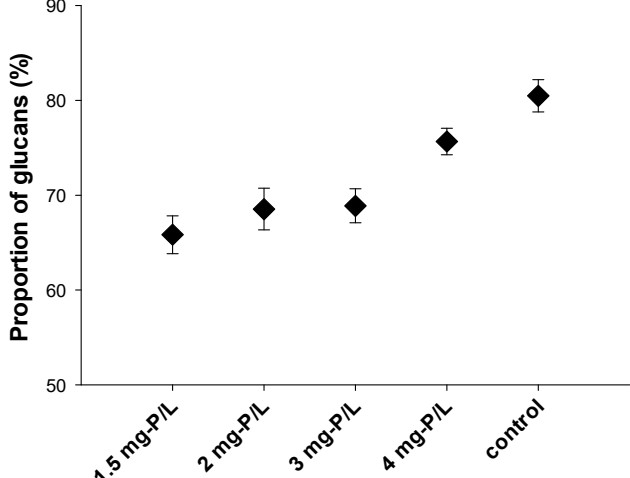

**Figure 3.** Proportion of total glucans in the total carbohydrate content in cultures of *A. platensis* grown with different phosphorus concentrations (mg-P/L). Results represents the average ± SD of *n* = 3 replicates.

So far, the storage polysaccharides of *A. platensis* are considered to be glycogen, which is comprised mainly of glucose units linked through 1,4-α glycosidic linkages and branched through 1,6-α glycosidic bonds [28,34–36]. However, more recently the study of Liu et al. [22] showed that after nitrogen starvation/limitation, *A. platensis* also produces a water-soluble polysaccharide that is composed mainly of 1.3:1.4- or 1.3:1.2-α glucans, which differs from glycogen. In the present study, it was demonstrated for the first time that besides α-glucans (glycogen), *A. platensis* also synthesize β-glucans to a high degree when phosphorus starvation/limitation is applied to the cultures.

β-glucans are polysaccharides consisting of D-glucose monomers linked by β-D-glycosidic bonds, which commonly exist in the cell walls of plants (especially cereals, such as oat barley and wheat), mushrooms, yeasts and bacteria but are absent in human cells. β-glucans from different sources have different structures, different degrees of branching and branching patterns as well as different molecular weights [37]. Cereal β-D-glucan, which consists of 1,3 and 1,4 linkages is reported to reduce blood glucose and cholesterol for obesity and cardiovascular disease (CVD),whereas yeasts, mushroom and bacteria β-D-glucan, which is characterized by 1,3 and 1,6 linkages [12], are proposed to have anti-tumor, anti-inflammation, and antiviral activities against immune system disease [10,11,38].

According to the U.S. Food and Drug Administration [39], regular consumption of 3 g/day of β-D-glucans reduces the risk of heart disease. *A. platensis* is a photosynthetic cyanobacterium belonging to the domain of bacteria; hence, it is not surprising that it synthesizes 1.3:1.6-β glucans. This is of particular interest because 1.3:1.6-β glucans display antitumor and immune stimulating properties; therefore, *A. platensis* enriched in β-glucans could potentially be used to develop functional foods [12].

### 3.3. Total Dietary Fiber

Figure 4 illustrates the content of total dietary fibers (TDF) in *A. platensis* in relation to the phosphorus available in the growth medium. The content of TDF was significantly enhanced (accounting for 20–32% of the dry biomass) under phosphorus limitation compared to the control (3.5%). TDF accounted for around 28–32% at 1.5–3 mg-P/L and decreased further to around 20% at the milder degree of phosphorus limitation (4 g-P/L). The levels of TDF were almost the same as the β-glucans and their ratios (TDF to β-glucans) ranged between 85% and 115%, suggesting that the main fraction of the TDF was most probably the accumulated β-glucans, as an insoluble fraction. However, this point deserves more in-depth studies in order to confirm this hypothesis. It is worth noting that the insoluble forms of β-glucans are less bioactive than the soluble ones [10]; therefore, it is probable that a chemical modification step would be required in order to increase their solubility and hence their bioactivity. Although *A. platensis* has attracted increasing interest as a protein source, there are no studies regarding the TDF content of the carbohydrate-enriched biomass of *A. platensis*. Typically, the TDF of *A. platensis* ranges between 3.6–8% [40–42]; however, the present study shows that as the carbohydrates content increases there is also a subsequent increase in TDF, which is also an interesting result given that dietary fibers are considered to have beneficial physiological effects on human health (such as laxation, lowering of blood cholesterol, diabetes control and digestive system improvements) [43,44].

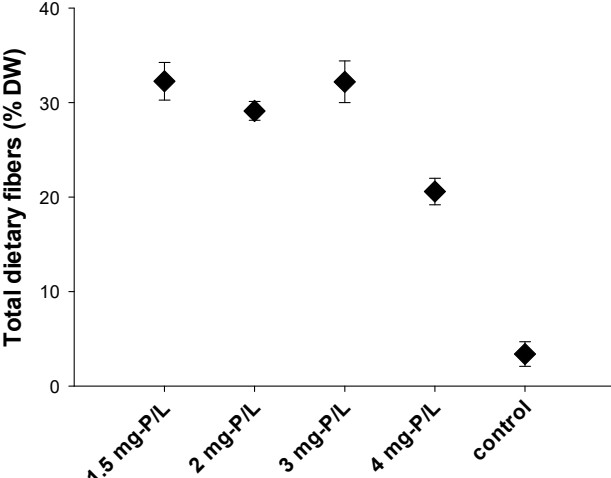

**Figure 4.** Total dietary fibers of *A. platensis* grown with different phosphorus concentrations (mg-P/L). Results represents the average ± SD of *n* = 3 replicates.

### 3.4. Monosaccharide Profile of A. platensis

Table 3 shows the main monosaccharides detected by HPLC after acid hydrolysis of *A. platensis* biomass. In all cases, the predominant monosaccharide was glucose, while galactose and mannose were also detected in minor amounts (Table 3). Note that mannose was not fully separated from citric acid, which could have contributed to these figures to some degree. The phosphorus limitation mainly affected the absolute amounts of glucose content, where as expected, glucose content was significantly enhanced at the lower end of phosphorus concentrations. The absolute amounts of galactose and mannose content were not affected in a clear way by phosphorus limitation. However, in terms of their ratio to

total sugars, phosphorus limitation did show a negative influence on galactose and mannose sugars. In general, these results are in line with other studies that also demonstrated that the predominant monosaccharide is glucose (more than 90%), with the presence of other monosaccharides in lesser amounts, such as galactose, rhamnose, arabinose, mannose, etc. [22,27,28]. It is worth noting that in all cases, the free monosaccharides (without acid hydrolysis) after cell lysis, and after treatment with ultrasonic waves (Transonic T460, 35 kHz for 1 h) represented less than 0.25% of the carbohydrates contained in the biomass. This indicates that the sugars contained in *A. platensis* were mainly polymers (polysaccharides).

**Table 3.** Monosaccharide profile of carbohydrates of *A. platensis* grown with different phosphorus concentrations. Results represent the average $\pm$ SD of *n* = 3 replicates.

| P-Concentration (mg-P/L) | Glucose (%DW) | Glucose (Ratio to Total Sugars, %) | Galactose (%DW) | Galactose (Ratio to Total Sugars, %) | Mannose (%DW) | Mannose (Ratio to Total Sugars, %) |
|---|---|---|---|---|---|---|
| 1.5 | 53.2 ± 0.50 | 97.3 | 0.6 ± 0.09 | 1.1 | 0.9 ± 0.01 | 1.6 |
| 2 | 55.3 ± 0.23 | 97.5 | 0.5 ±0.01 | 0.9 | 0.9 ± 0.01 | 1.6 |
| 3 | 43.9 ± 7.67 | 96.6 | 0.7 ± 0.08 | 1.6 | 0.8 ± 0.13 | 1.8 |
| 4 | 28.9 ± 0.38 | 94.9 | 0.9 ± 0.1 | 3.1 | 0.6 ± 0.07 | 2.0 |
| 89 (Control) | 24.6 ± 0.01 | 95.3 | 0.7 ± 0.02 | 2.8 | 0.5 ± 0.04 | 2.8 |

*3.5. FTIR Spectra*

Typical FTIR spectra from 4000 to 650 $cm^{-1}$ collected from *A. platensis* samples grown with different phosphorus concentrations are presented in Figure 5, and possible assignments of the main absorption bands are shown in Table 4. In brief, the main peaks with the highest absorbance observed for all the analyzed samples were found at 1080, 1015, 1005 (shoulder) $cm^{-1}$ (possibly from polysaccharides, carbohydrates, lipids and/or nucleic acids/phospholipids), at 1150 $cm^{-1}$ (from carbohydrates and/ or lipids), 1540 $cm^{-1}$ (mainly from proteins) and 1650 $cm^{-1}$ (mainly from proteins and/or olefinic and aromatic compounds) [45–49].

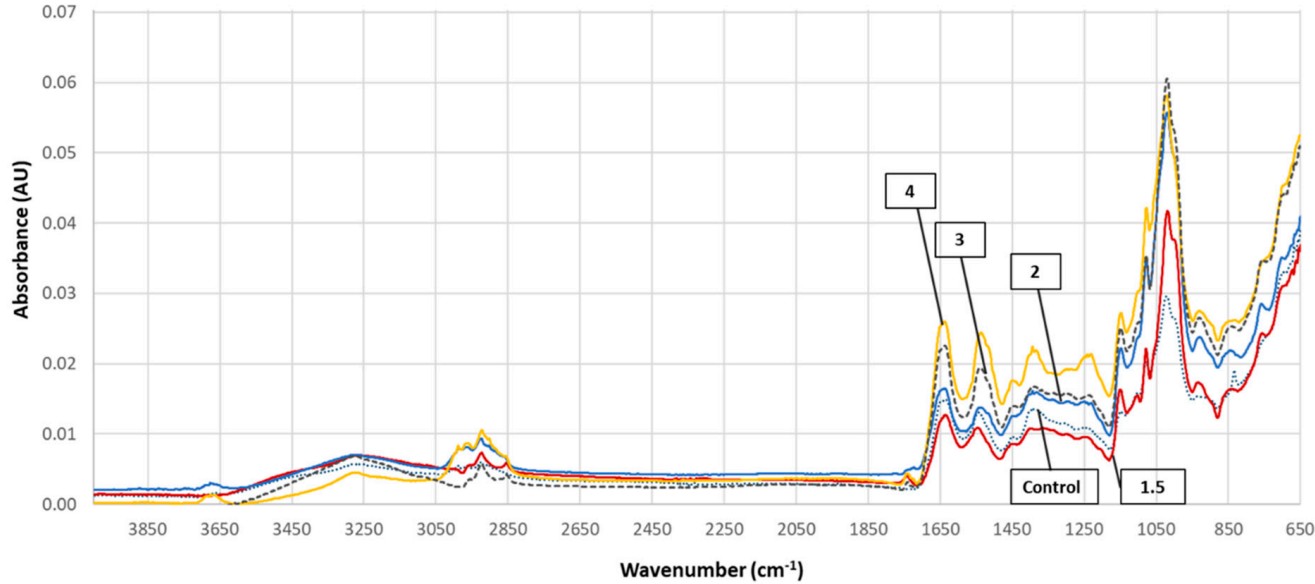

**Figure 5.** Raw FTIR spectra of A. platensis samples grown with different phosphorus concentrations: 1.5, 2, 3, 4, 89 (Control) mg-P/L.

**Table 4.** FT-IR frequencies observed in spectra collected from *A. platensis* samples and possible assignments of the vibration modes.

| Frequency (cm⁻¹) | Assignments | Reference |
|---|---|---|
| 3630–3040 (3280) [a] | N-H Stretch/protein (overtone of amide II band) | [45,48] |
| 3040–2800 (2860,2930, num [b]) | C–H symmetric stretch/CH2, CH3 lipid acyl chains, CH stretch/aminoacids or carbohydrates | [45–48] |
| 1775–1720 (1745) | C=O symmetric stretch/lipids, fatty acids | [45,48] |
| 1590–1715 (1650) | CO stretch, CN stretch, NH bend/amide I band (proteins), C=C stretch/olefinic and aromatic compounds | [45,47] |
| 1490–1590 (1540) | NH bend, CN stretch, CH stretch/amide II band (proteins) | [45,47,48] |
| 1490–1430 (1450) | CH3 and CH2 assymetric bend/proteins or lipids | [47–49] |
| 1430–1350 (1400) | CH3 and CH2 Symmetric bend/proteins or lipids, COO- symmetric stretch/acids | [47,49] |
| 1280–1185 (1240) | P = O asymmetric stretch/nucleic acids (DNA and RNA) or phosphorylated proteins, C-O-S/sulpholipids | [45,47,49] |
| 1185–1135 (1150) 1135–1095 (1105) | C-O-C stretch and O-H bend/carbohydrates or lipids | [45–47,49] |
| 1095–1070 (1080) 1070–950 (1015, 1005sh [c]) | C-O-H deformation, C-O-C stretch/polysaccharides, carbohydrates or lipids, P=O symmetric stretch/nucleic acids or phospholipids | [45,49] |
| 950–880 (940) | CH deformation/carbohydrates, P-O-P/polyphosphates | [45,48] |
| 880–825 (836-control samples only) | P-O assymetric stretch/lipids, CH deformation/carbohydrates, C-O-S/sulpholipids | [45] |
| 825–740 (760) | CH deformation/carbohydrates | [45] |
| 740–690 (690sh) | N-H bend/amide V band, CH2 rocking/lipids, CH2 rocking—NH deformation/polyglycines, O-CO/CO2/CO/C=O deformation/carboxylic acids or esters | [45] |

[a] main peak/peaks observed in the region, [b] numerous peaks, [c] shoulder.

Table 5 presents the performance indices for each PLS-R model built using the spectral data between the range of 1800–650 cm⁻¹, while Figure 6 presents the observed values vs. the model estimates for each studied parameter. It was observed that the indices were found to be satisfactory for all the estimated parameters except for the lipid content. In detail, after the external validation of the models (test set), the highest $R^2$ values were observed for the models built for glucose, dietary fibers, β-glucans, followed by carbohydrates and proteins, and finally by α-glucans, while for lipids a negative $R^2$ was observed. The same decreasing order was observed in the values of RPD. According to Nicolai et al. [50], the models for glucose, dietary fibers, β-glucans and carbohydrates showed high RPD values (above 3) that correspond to excellent prediction accuracy, the protein model's RPD value indicated good prediction accuracy (above 2.5), the values for the α-glucans indicated poor quantitative prediction (value between 2 and 2.5), while the RPD values for lipids model were very low (0.4). Regarding the RMSE values, the lowest values were observed for α- and β-glucans, followed by glucose, lipids and dietary fiber, and finally, by proteins and carbohydrates.

**Table 5.** Performance indices for the internal validation (LOO-CV) and the external validation (test set) estimates for each PLS-R model.

| | | Performance Indices | | | | | |
|---|---|---|---|---|---|---|---|
| Parameter | LV [a] | Internal Validation (LOOCV) | | | External Validation | | |
| | | R² [b] | RMSE [c] | RPD [d] | R² | RMSE | RPD |
| Proteins | 4 | 0.968 | 2.581 | 5.662 | 0.847 | 4.981 | 2.645 |
| Carbohydrates | 3 | 0.978 | 2.576 | 6.825 | 0.895 | 5.287 | 3.194 |
| Lipids | 2 | 0.440 | 1.054 | 1.360 | −5.449 | 2.255 | 0.408 |
| 1.3:1.6-β-glucans | 3 | 0.979 | 1.066 | 7.031 | 0.930 | 1.920 | 3.913 |
| α-glucans | 3 | 0.970 | 0.562 | 5.920 | 0.777 | 1.226 | 2.194 |
| Dietary fibers | 4 | 0.951 | 2.352 | 4.617 | 0.937 | 2.721 | 4.121 |
| Glucose | 4 | 0.921 | 3.630 | 3.623 | 0.973 | 2.053 | 6.286 |

[a] Number of latent variables used to calculate the PLS model, [b] Coefficient of determination, [c] Root mean square error, [d] Residual prediction deviation.

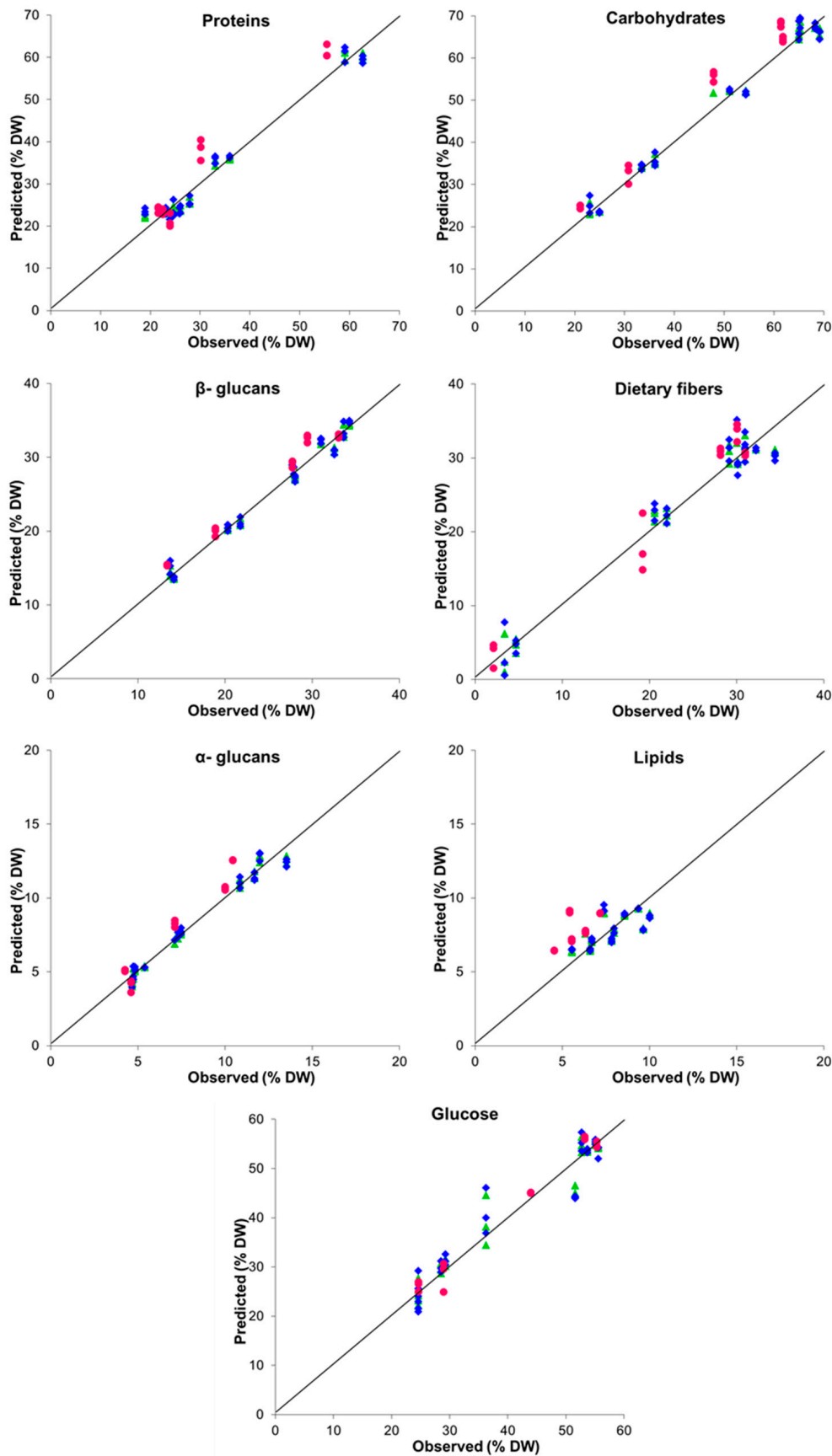

**Figure 6.** Comparison of the PLS-R–FTIR model estimates for the different *A. platensis* parameters against the experimentally observed values. ▲: calibration (training) data; ◆: internal validation (LOO-CV) data; ●: external validation (test set) data.

Previous studies have also included PLS-R modelling for the prediction of protein, carbohydrate and lipid in microalgae and cyanobacteria [51,52]. The study of Mayers et al. [51] showed good correlations for all the parameters studied in *Nannochloropsis* sp. between the predicted values using the FTIR spectra and the observed ones [51], while the study of Bataller and Capareda [52] also reported sufficient prediction models for the protein and carbohydrate content of Spirulina but insufficient models for lipids, a finding that was observed in this study as well. However, the aforementioned works applied one data set (one biological replicate/batch) to model the data and the models were not validated with an external independent data set (test set from a different batch), so the results cannot be fully compared with the current study, which included two batches for training and one for external validation. Finally, other studies have also evaluated the accuracy of the FTIR analytical method for the estimation of proteins, carbohydrates, lipids of microalgae and cyanobacteria (single batch cultures) and reported good correlations by plotting the actual values versus the predicted one or by fitting the data in a first order equation [46,47,49,53,54].

## 4. Conclusions

In the present study it is shown for the first time that the carbohydrates (polysaccharides) of A. platensis cultivated under phosphorus limitation consisted of a high degree of 1.3:1.6-β glucans, which accounted for around 20–34% of the dry biomass. It was also found that phosphorus limitation increased the TDF content of *A. platensis* from around 3.5% up to 32% (dry biomass). The PLS-R models developed using FTIR spectra of the different biomasses gave satisfactory predictions for the content of most biomass components (glucose, dietary fibers, β-glucans, followed by carbohydrates and proteins, and finally by α-glucans). Since β-glucans are of particular interest as biologically active compounds, this study suggests that *A. platensis* grown under phosphorus limitation could be a potential ingredient for the development of novel functional foods.

**Author Contributions:** Conceptualization, G.M. and D.A.; methodology, G.M., C.E., A.A.; formal analysis, G.M., A.A.; investigation, G.M., C.E., A.A., D.A.; resources, G.M., D.A.; data curation, G.M., C.E., A.A., D.A.; writing—original draft preparation, G.M., A.A.; writing—review and editing, G.M., C.E., A.A., D.A.; funding acquisition, G.M., D.A. All authors have read and agreed to the published version of the manuscript.

**Funding:** This study was funded by the Eranet BlueBio project AquaTech4Feed (General Secretariat for Research and Innovation GSRI, Greece, MIS 5070470/T11EPA4-00038).

**Institutional Review Board Statement:** Not applicable.

**Informed Consent Statement:** Not applicable.

**Acknowledgments:** Authors would like to thank Eleni Labrinea for conducting the HPLC analysis to determine the sugars.

**Conflicts of Interest:** The authors declare no conflict of interest.

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
