# Peer review of "Production of Arthrospira (Spirulina) platensis Enriched in β-Glucans through Phosphorus Limitation"

_applsci, doi:10.3390/app11178121_

Round 1

Reviewer 1 Report

The manuscript is covering two interesting aspects: sustainability and health. The improvement/steering options are clear, there could be a bit more discussion on the sustainability and health impact.

Detailed comments/suggestions:

lines 56-70: should include your own work (ref23) which is already showing the impact of P-limitation on CBH increase. First, this is from 2012, so I wonder if there is really no study done on P-limitation in those 9 yrs?!? very quick scan on pubmed showed several other publications (https://doi.org/10.3390/ijms16024250; https://doi.org/10.1002/btpr.2798) so what was lacking in these? and secondly, elaborate on what is new in the current study.

line 189: while the DW measurement is a fixed end-point (line 85: 30 days) one cannot mention growth rates, so please correct. Moreover, growth dynamics can be impacted to quite an extend due to P/nutrient-limitations (see ref3 of the current manuscript); hence it would be more insight full if DW development over the 30 days is shared...please add.

line 118: what is the amount of monosaccharides WITHOUT acid hydrolysis?

associated with the previous remark: the carbon-balance is not closed; respective numbers from the various figs and tables indicate for the lower P-samples for TDF (30%) --> glucans (45%) --> (glucose 55%) --> carbohydrates (65%); indicating quite some gaps, i.e. 10% more glucose than only in glucans (is there 10% free glucose on DM?), 10% more carbs than the total monosaccharides (table 3) indicate....hence, this must be discussed. Also, in view of my general starting remark: health! interesting that b-glucans are increased, but it should be clear what the other carbs are...more glucose is not persé healthy. Also, illustrated by Fig3...while total carbs is increasing, the relative percentage of glucans is decreasing with P-limitation...hence what are the other carbs formed/accumulated?

lines 221/22 and Fig2: the control is not following this trend: explain!

Author Response

The authors would like to thank Reviewer 1 for her/his positive feedback to our work. We revised the manuscript taking all comments into consideration. In what follows, we address point-to-point the comments which are also incorporated into the revised version of the manuscript.

The manuscript is covering two interesting aspects: sustainability and health. The improvement/steering options are clear, there could be a bit more discussion on the sustainability and health impact.

(Response: We added some new lines discussing more the aspects of sustainability and health impact of b-glucans production by A. platensis. Please see lines 32-36 and lines 261-274).

Detailed comments/suggestions:

lines 56-70: should include your own work (ref23) which is already showing the impact of P-limitation on CBH increase. First, this is from 2012, so I wonder if there is really no study done on P-limitation in those 9 yrs?!? very quick scan on pubmed showed several other publications (https://doi.org/10.3390/ijms16024250; https://doi.org/10.1002/btpr.2798) so what was lacking in these? and secondly, elaborate on what is new in the current study.

(Response: Indeed, there are some more recent publications on the topic of production of microalgal carbohydrates (we cite some additional publications in this point). We wanted however, to highlight in the original version of the manuscript that there is a lack of studies not on the production of carbohydrates per se but regarding the characterization of the produced carbohydrates. We revised this point in order to give clearer the contribution of the present study on the topic.)

line 189: while the DW measurement is a fixed end-point (line 85: 30 days) one cannot mention growth rates, so please correct. Moreover, growth dynamics can be impacted to quite an extend due to P/nutrient-limitations (see ref3 of the current manuscript); hence it would be more insight full if DW development over the 30 days is shared...please add.

(Response: thank you for the comment. We did remove the term “growth rates” and replaced with biomass concentration or biomass production. The cultivation mode was semi-continuous and there was a daily biomass harvest in a steady process without significant variations throughout the whole cultivation period as this was assured by the frequently monitoring of the biomass concentration through optical density. In any case any impact of the P-limitation would be shown in the biomass concentration of each culture since the process (after it reached a steady state) was maintained for more than 30 days.

We added the following text in the revised manuscript: “The cultures were frequently (every 3-4 days) monitored through optical density (OD@750nm) to ensure that the process was steady in terms of biomass concentration. The values of OD did not vary throughout the cultivation period more than 10%.”.).

line 118: what is the amount of monosaccharides WITHOUT acid hydrolysis?

(Response: Since we wanted to see the variation of the total monosaccharides profile in relation to the P-limitation degree, we did only conduct HPLC after biomass hydrolysis in order to get the total amount of monosaccharides. Any possible free monosaccharides (without hydrolysis) contained in the biomass would have been included in the hydrolysate.). However, based on the comment we did some additional measurements on the free monosaccharides (measured as reducing sugars with the DNS method) after cell lysis in a sonic bath (1 hour). The amount of free sugars was very low (less than 0.25% in all cases). We now included an additional sentence, which reads: “It is worth noting that the free monosaccharides (without acid hydrolysis) after cell lysis after treatment with ultrasonic waves (Transonic T460, 35 kHz for 1 hour), in all cases, represented less than 0.25% of the carbohydrates contained in the biomass. This indicates that sugars contained in A. platensis were mainly as polymers (polysaccharides)”.]

associated with the previous remark: the carbon-balance is not closed; respective numbers from the various figs and tables indicate for the lower P-samples for TDF (30%) --> glucans (45%) --> (glucose 55%) --> carbohydrates (65%); indicating quite some gaps, i.e. 10% more glucose than only in glucans (is there 10% free glucose on DM?), 10% more carbs than the total monosaccharides (table 3) indicate....hence, this must be discussed.

Also, in view of my general starting remark: health! interesting that b-glucans are increased, but it should be clear what the other carbs are...more glucose is not persé healthy. Also, illustrated by Fig3...while total carbs is increasing, the relative percentage of glucans is decreasing with P-limitation...hence what are the other carbs formed/accumulated?

(Response: thank you very much for this interesting comment. Indeed, in all cases the different fractions measured were propositionally lower compared to each of their broader category. This could be partially explained by two reasons. (1) most probably because the broader categories might contain some other types of biomolecules that cannot be assayed during the specific analysis and (2) due to some limitations derived from analytical processes because of pontential low recoveries during the analysis etc.

lines 221/22 and Fig2: the control is not following this trend: explain!

[Response: Thank you very much for giving us the opportunity to explain this point. We added the following piece in the revised manuscript: “However, the ratio β/α- glucans of the control culture did not follow this trend indicating that there was an unbalanced accumulation of α- and β-glucans. Given that the α-glucans content of the cultures with 4 mg-P/L was almost the same as the control cultures, this might show that under moderate P limitation degrees first β-glucans start to accumulate and then as the P limitation degree increases then α-glucans start also to increase in an unbalanced increase pattern between α- and β- glucans.

Reviewer 2 Report

This is an interesting study about production of Arthrospira platensis enriched in β-glucans through phosphorus limitation. The manuscript itself is well written. Authors put a large amount of work into it, which is appreciated. While an entire process might be of potential use in some countries, some minor criticism can be drawn:

Introduction

L29-L46 - in this section authors are stating solely the good sides of Spirulina plant, yet not much drawbacks came into attention - this might be a very potent source of heavy metals and Fluoride, which are vastly different in terms of toxicity and enviromental hazard. Please incorporate and cite at least publons.com/p/36379569/ and https://pubmed.ncbi.nlm.nih.gov/32504425/ with https://pubmed.ncbi.nlm.nih.gov/24235875/ followed.

Materials and methods

L79-80 'for agitation reasons' is vague, either remove it or use more convenient term

L83 - light cycle 16/8 requires relevant citation

L91 - filter paper manufacturer data is missing

L105 - Lowry et al. - there is no need for Italics

L118 - HPLC abbreviation is used here fot the first time, however there is no explanation - please provide, according to MDPI abbreviations policy

L124 - Aminex manufacturer data is missing

L125 - Differential refractometer manufacturer data is missing

L128-129 - please provide relevant data for software used

Author Response

We would like to thank Reviewer #2 for her/his positive comments on our work. We have revised the manuscript addressing all the comments.

This is an interesting study about production of Arthrospira platensis enriched in β-glucans through phosphorus limitation. The manuscript itself is well written. Authors put a large amount of work into it, which is appreciated. While an entire process might be of potential use in some countries, some minor criticism can be drawn:

Introduction

L29-L46 - in this section authors are stating solely the good sides of Spirulina plant, yet not much drawbacks came into attention - this might be a very potent source of heavy metals and Fluoride, which are vastly different in terms of toxicity and enviromental hazard. Please incorporate and cite at least publons.com/p/36379569/ and https://pubmed.ncbi.nlm.nih.gov/32504425/ with https://pubmed.ncbi.nlm.nih.gov/24235875/ followed.

[Response: thank you for the comment. Indeed, there were some reports that showed that Spirulina can be a source of heavy metals due to its ability of hyperaccumulation. However, this is true only when Spirulina is produced with seawater or harvested from natural water bodies (like lakes etc.) that might be contaminated by heavy metals. In our study we demonstrate a process that deals with the cultivation under controlled environment (including of good quality of water to form the growth medium). Al-Dhabi 2013 examined 25 commercialized food samples of Spirulina for their content of Ni, Zn, Hg, Pt, Mg and Mn. He found that all studied samples were within the daily uptake levels, so their consumption was con-sidered as safe (Al-Dhabi 2013).

Reference: Al-Dhabi, N.A. Heavy metals analysis in commercial Spirulina products for human consumption. Saudi J. Biol. Sci. 2013, 20, 383-388]

Materials and methods

L79-80 'for agitation reasons' is vague, either remove it or use more convenient term

[Response: The point has been revised. Now it reads: “The cultures were aerated with filtered air provided by a membrane air pump to agitate the culture broth and to keep the cells in suspension”.

L83 - light cycle 16/8 requires relevant citation

[Response: the light cycle 16/8 (day/night) was chosen after some preliminary trials, where it was observed that longer light exposure increases carbohydrate content in Spirulina.]

L91 - filter paper manufacturer data is missing

[Response: we have now specified the manufacturer and the grade of the filter paper used)

L105 - Lowry et al. - there is no need for Italics

[Response: corrected]

L118 - HPLC abbreviation is used here fot the first time, however there is no explanation - please provide, according to MDPI abbreviations policy

[Response: corrected]

L124 - Aminex manufacturer data is missing

[Response: corrected]

L125 - Differential refractometer manufacturer data is missing

[Response: corrected]

L128-129 - please provide relevant data for software used

[Response: data were processed with Agilent ChemStation Software].

Reviewer 3 Report

Line 32: They have gained interest as cell factories because of their advantages over traditional practices employing heterotrophs [1]. What are the advantages over traditional practices employing heterotrophs?

Line 35: Microalgae and cyanobacteria synthesize carbohydrates through photo- synthesis and carbon fixation processes, which have diverse structural (cell-walls etc.) and metabolic (energy storage etc.) roles. Suggest to divide this sentence into two sentences.

Line 42: No reference for biological effects? Suggest review papers such as: DOI: 10.3390/molecules24071251; DOI: 10.3390/jof6040356.

Line 54: Are there any studies on beta-glucans from this source?

The method section needs to be divided up per assay, instead of one big long section. Eg: 2.3 FTIR Analysis, 2.4 HPLC Analysis of…

Figure 1: What does P/L mean on the x axis?- should say in legend?

Figure 2: As beta-glucans are the main focus of this work it would be nice to see them in a graph alone instead of ratio?

Figure 4: Please put a space between % and D

Figure 6: Excellent graphs but not very clear

Author Response

We would like to thank Reviewer #3 for her/his time for reviewing and commenting on our manuscript. We have revised the manuscript addressing all the criticism raised by the reviewer.

Line 32: They have gained interest as cell factories because of their advantages over traditional practices employing heterotrophs [1]. What are the advantages over traditional practices employing heterotrophs?

[Response: We have revised this point in order to avoid raising any possible debate. The sentence now reads: They have gained interest as cell factories as alternative pathway to the traditional practices employing heterotrophs [1]].

Line 35: Microalgae and cyanobacteria synthesize carbohydrates through photo- synthesis and carbon fixation processes, which have diverse structural (cell-walls etc.) and metabolic (energy storage etc.) roles. Suggest to divide this sentence into two sentences.

[Response: We now divided it into two sentences].

Line 42: No reference for biological effects? Suggest review papers such as: DOI: 10.3390/molecules24071251; DOI: 10.3390/jof6040356.

[Response: We now added the suggested review paper as a reference].

Line 54: Are there any studies on beta-glucans from this source?

[Response: We have now added the following sentence: “However, there are no studies known so far dealing with β-glucans produced by A. platensis.”.

The method section needs to be divided up per assay, instead of one big long section. Eg: 2.3 FTIR Analysis, 2.4 HPLC Analysis of…

[Response: We now divided the section as suggested by the reviewer].

Figure 1: What does P/L mean on the x axis?- should say in legend?

[Response: it denotes the phosphorus concentration mg-P/L (mg phosphorus per liter). We now included it also in the legend in order to facilitate the readers].

Figure 2: As beta-glucans are the main focus of this work it would be nice to see them in a graph alone instead of ratio?

[Response: the content of β- and α-glucans are shown in Table 2 and any display in a figure will be a repetition that we should avoid].

Figure 4: Please put a space between % and D

[Response: done]

Figure 6: Excellent graphs but not very clear

[Response: we now improved the figure for more clarity]